# Playing Board Games with the Predict Results of Beam Search Algorithm

## Abstract

This paper introduces a novel algorithm for two-player deterministic games with perfect information, which we call PROBS (Predict Results of Beam Search). Unlike existing methods that predominantly rely on Monte Carlo Tree Search (MCTS) for decision processes, our approach leverages a simpler beam search algorithm. We evaluate the performance of our algorithm across a selection of board games, where it consistently demonstrates an increased winning ratio against baseline opponents. A key result of this study is that the PROBS algorithm operates effectively, even when the beam search depth is considerably smaller than the average number of turns in the game.

## 1 Introduction

In the domain of artificial intelligence, two-player board games have historically served as pivotal 'toy problems' for exploring and advancing search and planning algorithms within vast decision spaces. The outstanding algorithm AlphaZero (Silver et al. (2016) Silver et al. (2017a) Silver et al. (2017b)) achieved superhuman performance in the game of Go, chess, and other board games without the use of human expertise in these games. In this work, we introduce a new approach to solving such games. The main idea is that the algorithm iterates through possible moves using beam search, and then learns to predict the outcome of this search. This concept gives rise to the name of the algorithm, PROBS - Predict Results of Beam Search. This approach shows promising results — it demonstrates an increase in the winning percentage during the training process and shows improvement with the use of greater computational power. Although this new approach to solving board games does not improve upon state-of-the-art approaches, it demonstrates a new working concept that may inspire researchers to develop new methods in other areas.

The foundation of the PROBS algorithm is the iterative training of two neural networks. The first network is a value function, $V(s)$, which predicts the expected utility from the current state. $V(s)$ approximates the optimal value function $V^*(s)$, which exists for all games of perfect information and determines the outcome of the game under perfect play by all players.

The agent's action selection is modeled by a second network, $Q(s, a)$, which predicts the outcome of a beam search in the game sub-tree from state $s$ with move $a$. Conducting a full traversal of the entire game tree from state $s$ is unfeasible; therefore, the algorithm only iterates over a limited subtree and replaces the values of the leaves of this tree with $V(\text{leaf\_state})$.

These two neural networks are trained iteratively through a cycle of the following steps:

1. The agent plays games against itself using $Q(s, a)$, selecting both optimal and suboptimal moves with a certain probability to ensure exploration.

2. Using the games played, the agent trains the value function $V(s)$, which predicts the expected utility of being in state $s$ with the policy derived from applying $Q(s, a)$.

3. For each observed state from the same played games, a beam search (Lowerre (1976)) is initiated to explore a subtree, utilizing a fixed version of $Q(s, a)$ to prioritize the expansion of each state.

When the search limit is reached, the value of the leaf states is replaced using $V$. The results of this exploration are used to improve the function $Q(s, a)$.

Each iteration guides the policy represented by $Q(s, a)$ towards one where decisions are made by exploring a small game sub-tree, and the leaves of this tree are replaced with $V(s)$. Thus, $Q(s, a)$ becomes a slightly deeper estimation compared to the function $V(s)$, and $V(s)$ in turn becomes an estimate of this new version of $Q(s, a)$. In the subsequent iteration, $Q(s, a)$ is trained on game sub-trees where the values of the leaves are replaced with this more accurate version of $V(s)$. As a result, in each iteration, $Q(s, a)$ begins to incorporate information about good moves deeper in the game sub-tree than the depth of the beam search.

The PROBS algorithm can be intuitively understood by comparing it to how chess players improve their game. Initially, the functions $Q(s, a)$ and $V(s)$ are randomly initialized, so in the very first step of the first iteration, the agent plays randomly. However, the first iteration of training $V(s)$ can already begin to form some understanding of the game — it might recognize that material advantage leads to victory, and that checks and threats against strong pieces are also advantageous. Then, like chess players who "calculate the best move in a position", the agent begins to ponder each of its moves. Similar to chess players, for each position, the agent initiates a search for its possible best moves and those of the opponent. As a chess player spends many hours contemplating their moves, they learn to improve the process of calculation itself — focusing more on better moves and seeing benefits even before calculating all combinations.

## 2 Related Work

Outstanding success in board games was achieved by AlphaZero, which reached a superhuman level in the game of Go. Similar to our work, AlphaZero iteratively optimizes both $V(s)$ and $Q(s, a)$; however, the optimization of $Q(s, a)$ is achieved through the use of a Monte Carlo Tree Search (MCTS) tree, the outcomes of which are used as estimates of move probabilities. These probabilities serve as the target for training $Q(s, a)$. The more moves the agent evaluates while creating the MCTS tree, the more accurately these probabilities are estimated. In contrast, the PROBS algorithm does not evaluate move probabilities but rather assesses the outcomes of tree exploration. Consequently, in PROBS, there is no MCTS tree but a simpler beam search mechanism is used instead. In this work, we demonstrate that even traversal of an extremely small subtree allows each iteration to enhance the policy, showing that limited yet focused exploration can effectively contribute to strategy refinement in deterministic games with perfect information.

In addition to board game strategies, advancements in planning algorithms have been explored in the context of puzzle games, such as demonstrated in "Beyond A*: Better Planning with Transformers via Search Dynamics Bootstrapping" (Lehnert et al. (2024)). This study focuses on puzzles like Sokoban (Wikipedia (2024d)), where the authors predict the entire path from the initial state to a goal state using A*. The fundamental difference between board games and puzzle games lies in the nature of the objectives. In puzzle games, the task is to solve the puzzle, and all paths that solve the puzzle are equally valid. However, board games involve two players with opposing goals, making the objective to develop a policy that remains unbeaten by any player. Any discovered path might be dominated by another, rendering the board game scenario a moving target problem, where A* is not directly applicable.

The paper "Offline Reinforcement Learning as One Big Sequence Modeling Problem" (Janner et al. (2021)) utilizes beam search to generate action sequences that maximize rewards, employing a transformer architecture for this purpose. Similar to the findings in Lehnert et al. (2024), they also rely on a well-defined notion of effectiveness for action sequences. They demonstrate that generalizing from beam search results can yield effective strategies, if objectives are clear. In our study, we illustrate that generalization from beam search can lead to iterative improvements in the strategies discovered and is not myopic.

## 3 The PROBS Algorithm

Our method uses two independent deep neural networks:

- $V_\theta(s)$, parameterized by $\theta$, takes the raw board position $s$ as input and produces its value. The value of a terminal state is 1, -1, or 0, reflecting a win, loss, or draw, respectively. The output of the value model is a number between -1 and 1, representing the long-term expected reward from following a policy derived from $Q_\phi(s, a)$.

- The network $Q_\phi(s, a)$, with parameters $\phi$, takes the raw board position $s$ as input and produces a vector of q-values. By applying a softmax function to the predicted q-values, we derive action probabilities, thereby enabling $Q_\phi$ to represent the policy of a trained agent. This vector of values represents the outcome of a beam search used to select an appropriate action from the state $s$.

The training of these two networks follows an iterative process, starting with self-play using the $Q_\phi$ model. This is followed by refining $V_\theta$ using the outcomes of the played games, and then enhancing $Q_\phi$ with beam search. Every iteration in the process acts as an improvement operator for the policy encoded by $Q_\phi$. The following is a more detailed overview of each iteration:

- Execute a predefined number of self-play games, selecting moves based on the q-values derived from $Q_\phi(s, a)$. Action probabilities are obtained by applying the softmax function to the vector of q-values, followed by the selection of a random action using these probabilities. To increase exploration, Dirichlet noise is added to the action probabilities, following the approach outlined in Silver et al. (2017b), with parameters $\varepsilon = 0.25$ and $\alpha$ tailored to each game:

$$p_a = \frac{e^{Q_\phi(s,a)}}{\sum e^{Q_\phi(s,*)}}$$
$$P(s, a) = (1 - \varepsilon)p_a + \varepsilon\eta_a$$
$$\eta_a \sim \text{Dir}(\alpha)$$

- Parameters $\theta$ of the value model $V_\theta(s)$ are optimized via gradient descent, with a loss function that computes the mean-squared error between the predicted and actual terminal rewards at the conclusion of each game episode, with each state $s$ being drawn randomly from an experience replay.

- In each observed state $s$, we deploy a beam search to generate a limited sub-tree of the game, starting from $s$. The breadth and depth of this sub-tree are important parameters of our model. The leaf states of this sub-tree are either terminal states or the limits of tree expansion. Values for terminal leaf states are provided by the emulator, typically set to 1, 0, or -1; values at the limits of the beam search game sub-tree are estimated using $V_\theta(s)$. The value of any non-leaf state is the maximum of the negative values of its child nodes.

- The parameters $\phi$ of the q-value model $Q_\phi(s, a)$ are optimized via gradient descent, with a loss function that computes the mean-squared error between the predicted q-values for each valid action $a$ within a state, and the corresponding q-values obtained through beam search from state $s$ upon selecting action $a$.

- Clear the experience replay buffer.

The following pages provide detailed descriptions of the beam search and the PROBS algorithm.

---

**Algorithm 1** Beam Search

---

**Data:**
    Board state $s$
    State value estimator function $V_\theta$
    Q-values estimator function $Q_\phi$
    Number of nodes to expand in game sub-tree $E$
    Max depth of game sub-tree $M$
**Returns:** Q-values for every valid action in $s_0$

  1: **function** BEAMSEARCH($s, V_\theta, Q_\phi, E, M$)
  2:    $tree \leftarrow$ empty list
  3:    $beam \leftarrow$ empty priority queue
  4:    $tree.\text{add}(\{value = \emptyset, state = s, children = \emptyset\})$
  5:    $beam.\text{add}(\{priority = \infty, nodeIndex = 1, depth = 0\})$
  6:    **for** $expand = 1, E$ **do**
  7:        **if** $beam$ is empty **then**
  8:            **end for**
  9:        **end if**
10:        $priority, nodeIndex, depth \leftarrow$ pop item with the highest priority from $beam$
11:        $value, state, children \leftarrow tree[nodeIndex]$
12:        $actionValues \leftarrow Q_\phi(state)$
13:        **for all** $action \in emulator.\text{getValidActions}(state)$ **do**
14:            $nextState, reward, done \leftarrow emulator.\text{step}(state, action)$
15:            $childIndex \leftarrow$ length of tree $+ 1$
16:            $children.\text{add}(\{action = action, child = childIndex\})$
17:            **if** $done$ **then**
18:                $tree.\text{add}(\{value = -reward, state = nextState, children = \emptyset\})$
19:            **else**
20:                $tree.\text{add}(\{value = \emptyset, state = nextState, children = \emptyset\})$
21:                **if** $depth < M$ **then**
22:                    $priority \leftarrow (\infty$ if $depth = 0$ else $actionValues[action])$
23:                    $beam.\text{add}(\{priority, childIndex, depth + 1\})$
24:                **end if**
25:            **end if**
26:        **end for**
27:    **end for**
28:    **for** $i$ in range from length of tree to 1 **do**
29:        $value, state, children \leftarrow tree[nodeIndex]$
30:        **if** $value = \emptyset$ **then**
31:            **if** $children$ is empty **then**
32:                $tree[i].value \leftarrow V_\theta(state)$
33:            **else**
34:                $tree[i].value \leftarrow \max(-tree[child].value$ for $child$ in $children)$
35:            **end if**
36:        **end if**
37:    **end for**
38:    $QValues \leftarrow (action, -tree[child].value)$ for $(action, child)$ in $tree[1].children$
39:    **return** $QValues$
40: **end function**

---

---

**Algorithm 2** PROBS - Predict Results of Beam Search

---

**Data:**

    $N_{ITER}$ - number of iterations

    $N_{EPISODES}$ - number of episodes to play in each iteration

    $N_{TURNS}$ - number of maximum turns to play in each episode

    $C$ - capacity of the experience replay memory

    $E$ - number of expanded nodes in game sub-tree for each beam search

    $M$ - max depth of game sub-tree for each beam search

    $\varepsilon$ - exploration coefficient

    $\alpha$ - Dirichlet noise parameter to boost exploration

**Returns:** $Q_\phi$ - q-values estimator function

1: **function** PROBS($E, M, C, N_{ITER}, N_{EPISODES}, N_{TURNS}, \varepsilon, \alpha$)

2:     Initialize experience replay memory $D_{ER}$ to capacity $C$

3:     Initialize value function $V_\theta$ with random weights $\theta$

4:     Initialize q-value function $Q_\phi$ with random weights $\phi$

5:     **for** iteration = 1, $N_{ITER}$ **do**

6:         **for** episode = 1, $N_{EPISODES}$ **do**

7:             Reset environment emulator and observe initial state $s_0$

8:             Store $s_0$ in $D_{ER}$ as a beginning of a new episode

9:             **for** t = 1, $N_{TURNS}$ **do**

10:                 Compute action probabilities using:

$$p_a = \frac{e^{Q(s_t, a)}}{\sum_{a'} e^{Q(s_t, a')}}$$

$$P(s_t, a) = (1 - \varepsilon)p_a + \varepsilon\eta_a$$

$$\eta_a \sim Dir(\alpha)$$

11:                 Draw a random valid action $a$, using probabilities $p_a$

12:                 Execute action $a$ in emulator and observe reward $r_t$ and state $s_{t+1}$

13:                 Store $s_{t+1}$ in $D_{ER}$

14:                 End loop if environment is terminated

15:             **end for**

16:             Associate final reward $r_T$ with all the states in the episode: $(s_t, \delta_t r_T)$, where $\delta_T = 1; \delta_{T-1} = -1; \delta_{T-2} = 1; \delta_{T-3} = -1$ and so on.

17:         **end for**

18:         Initialize dataset $D_V$ as empty and put all the observed pairs $(s_t, \delta_t r_T)$ into it

19:         **for all** random minibatch in $D_V$ **do**

20:             Perform a gradient step on $(\delta_t r_T - V(s_t; \theta))^2$ with respect to the network V parameters $\theta$

21:         **end for**

22:         Initialize dataset $D_Q$ as empty

23:         **for all** state $s_t$ in $D_{ER}$ **do**

24:             $QValues \leftarrow beamSearch(s_t, V_\theta, Q, E, M)$

25:             Put $(s_t, QValues)$ into dataset $D_Q$

26:         **end for**

27:         **for all** random minibatch in $D_Q$ **do**

28:             Perform a gradient step on $(QValues[a] - Q(s_t, a; \phi))^2$ with respect to the network Q parameters $\phi$

29:         **end for**

30:         Clear experience replay $D_{ER}$

31:     **end for**

32:     **return** $Q_\phi$

33: **end function**

---

## 4    Empirical evaluation

We evaluate the PROBS algorithm on the game of Connect Four (Wikipedia (2024a)), a classic two-player deterministic game with perfect information, featuring a board size of 6x7 and a maximum of 7 actions per turn. The algorithm was compared against four distinct agents:

- Random agent, which performs any valid move at random.

- One-step lookahead agent, which analyzes all potential moves to either execute a winning move, if available, avoid immediate losing moves, or otherwise select randomly from the remaining moves.

- Two-step lookahead agent that evaluates the game tree up to two moves ahead with similar decision criteria.

- Three-step lookahead agent that extends this evaluation to three moves ahead, maintaining the same strategic approach.

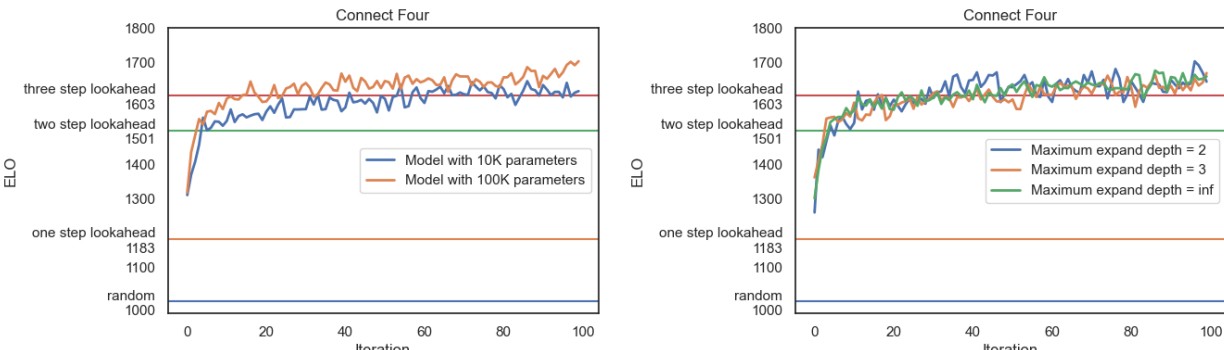

Figure 1: (left) Training the PROBS algorithm on the Connect Four board game using various model sizes. (right) Training the PROBS algorithm with varying depth limits for beam search.

In our experiment, the average game lasted for 19 turns, which makes Three-step lookahead agent quite effective. To illustrate the learning progression of the PROBS algorithm, we evaluated each iteration checkpoint against these four agents and reported its Elo rating (Elo (1966)). Before the experiment, we determined the Elo ratings of these four agents, using the Wikipedia (2024b), to be 1000, 1183, 1501, and 1603, respectively.

Figure 1 (left) illustrates the outcome of the training process for two different models. It shows two lines, each representing the mean Elo rating for every iteration, aggregated over multiple parallel training runs initiated from scratch under different parameter settings. Specifically, each training run encompassed 100 iterations, involving 1,000 games per iteration. The training runs varied in terms of node expansions (10, 30, or 100) and the maximum depth allowed for beam search (2, 3, or 100).

Figure 1 (right) demonstrates that the PROBS algorithm performs effectively with various depth limit values for beam search. Notably, even with beam search constrained to a maximum depth of 2, the PROBS algorithm can be trained to win significantly against a "three-step lookahead" agent (Elo rating 1603), which performs a full scan of all actions for the game sub-tree at depth 3. It is generally improbable for a player trained only up to a depth of 2 to defeat a player who performs optimally with a depth of 3 search, unless it can leverage information beyond this 3-step lookahead. This suggests that during its iterative training process, the PROBS agent learns to utilize information exceeding its beam search constraints.

We also trained the PROBS algorithm on computationally more challenging games, as shown in 2. Toguz-Kumalak (Wikipedia (2024e)), a two player game, has its board state encoded in two pairs of tensors: an 18x84 tensor for the board and a 2x9 tensor for the kazna, totaling 1530 inputs. The player can choose from 9 actions at each turn. We employed a beam search strategy with a maximum of 50 node expansions and a

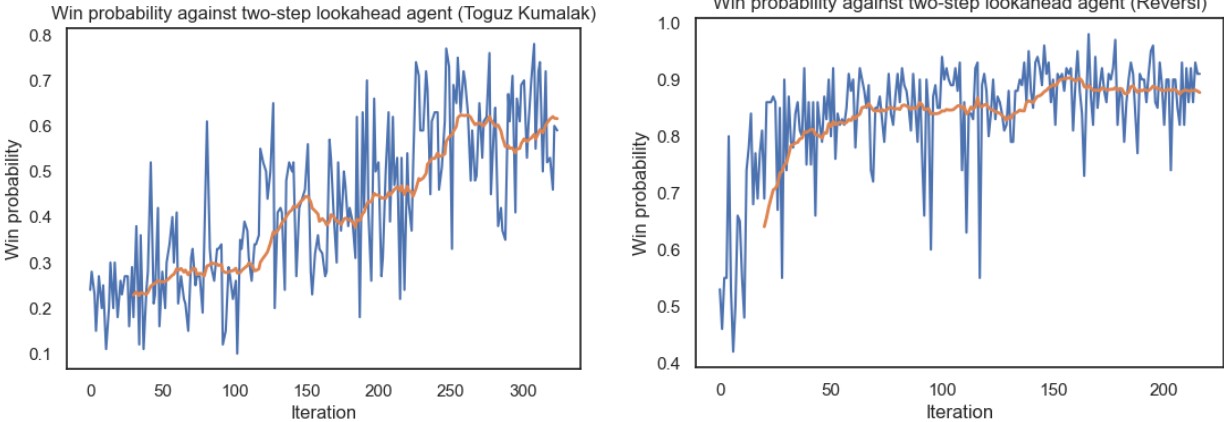

Figure 2: Training PROBS algorithm on Toguz-Kumalak and Reversi (Othello)

depth limit of five. Each iteration had 200 games. On average, training sessions for Toguz Kumalak lasted for 92 steps, with a maximum turn cap of 100.

Another game tested was Reversi (Wikipedia (2024c)), using the Othello variation, where the board is encoded with a 4x8x8 tensor and the player has 65 actions to choose from at each turn. Here, we used a beam search with a maximum of 500 node expansions and a depth limit of 5. Each iteration had 200 games. During training, Reversi games averaged 61 turns, under the same maximum turn cap of 100.

## 5 Conclusion, limitations and future work

In this work, we introduce a novel algorithm, PROBS, which leverages a combination of deep neural networks and beam search, consistently demonstrating an increased winning ratio against baseline opponents. We had shown that the PROBS algorithm, when applied with a limited beam search, progressively improves throughout self-play iterations and consistently winning against a model which performs a full scan of actions in a deeper sub-tree.

Due to computational constraints, we were unable to directly compare the PROBS algorithm with its main competitor, Alpha-Zero. Since implementations of Alpha-Zero are highly optimized and trained on large clusters, a direct comparison with our novel algorithm would not be fair. The goal of this paper is to introduce PROBS and showcase its potential.

Future work should also consider applying the core ideas of the algorithm to broader problems such as imperfect information games, continuous action spaces, and non-deterministic games. We strongly believe that integrating deep neural network capabilities with classic graph search algorithms holds significant potential.

## 6 Configuration

We used framework OpenSpiel (Lanctot et al. (2019)) for environment emulators. We used the following settings for each game:

- $N_{ITER}$ - number of iterations
- $N_{EPISODES}$ - number of episodes to play in each iteration
- $N_{TURNS}$ - number of maximum turns to play in each episode
- $E$ - number of expanded nodes in game sub-tree for each beam search
- $M$ - max depth of game sub-tree for each beam search

- $C$ - capacity of the experience replay memory

- $\varepsilon$ - exploration coefficient

- $\alpha$ - Dirichlet noise parameter to boost exploration

**Connect Four:** $N_{\text{ITER}} = 100, N_{\text{EPISODES}} = 1000, N_{\text{TURNS}} = 100, E$ values (10, 30, 100), $M$ values (2, 3, 99), $C = 1e5, \varepsilon = 0.25, \alpha = 0.5$, learning rate 0.003 for both models, batch size 128. Networks $V$ and $Q$ consist of convolutions and dense layers, with leaky relu (0.01) for activation. We experimented with two model sizes: smaller networks of 4 layers (10K parameters) and larger networks of 5 layers (100K parameters).

**Toguz-Kumalak:** $N_{\text{ITER}} = 326, N_{\text{EPISODES}} = 200, N_{\text{TURNS}} = 100, E = 50, M = 5, C = 1e5, \varepsilon = 0.25, \alpha = 0.2$, learning rate 0.0003 for both models, batch size 128. Networks $V$ and $Q$ of the same structure, 7 layers, 420K parameters.

**Reversi:** $N_{\text{ITER}} = 200, N_{\text{EPISODES}} = 200, N_{\text{TURNS}} = 100, E = 500, M = 5, C = 1e5, \varepsilon = 0.25, \alpha = 0.2$, learning rate 0.001 for both models, batch size 128. Networks $V$ and $Q$ of the same structure, 5 layers, 230K parameters.

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
