# OpenReview forum: "Playing Board Games with the Predict Results of Beam Search Algorithm"
_TMLR — Rejected by TMLR_

### Review · Reviewer_wXDx · 2024-10-06

**Summary Of Contributions:**

This paper introduces the PROBS (Predict Results of Beam Search) algorithm for two-player deterministic games with perfect information. PROBS leverages beam search alongside two neural networks: a value network V(s) predicting the expected utility from a state s, and a q-value network Q(s,a) estimating the outcomes of beam searches from s with action a. Through iterative self-play and training, the algorithm aims to improve the policy encoded in Q(s,a), even when the beam search depth is significantly less than the average game length. The authors evaluate PROBS on games like Connect Four, Toguz-Kumalak, and Reversi, reporting increased winning ratios against baseline opponents.

**Audience:**

No

**Broader Impact Concerns:**

The paper does not raise immediate ethical or societal concerns.

**Claims And Evidence:**

No

**Requested Changes:**

- **Include Comparisons with State-of-the-Art Methods:**
    - Provide experimental results comparing PROBS with strong baselines like AlphaZero or its variants, even on smaller games if computational resources are limited.
    - Discuss the algorithm's performance relative to these methods, highlighting strengths and weaknesses.
- **Discuss Relevant Literature:**
    - Expand the related work section to include key papers that combine search and learning, such as Anthony et al. (2017) and Guez et al. (2018), which explore integrating planning with learning.
    - Explain how PROBS differs from and builds upon these methods.
- **Strengthen Experimental Evaluation:**
    - Test PROBS against stronger opponents, such as agents using deeper search depths or MCTS-based methods, to better assess its capabilities.
    - Include ablation studies to understand the impact of beam search depth, number of node expansions, and network architectures on performance.
    - Analyze sensitivity to hyperparameters.
- **Provide Theoretical Insights:**
    - Add a theoretical discussion on why PROBS is expected to perform well, potentially relating it to policy iteration or value iteration frameworks.
    - Discuss any convergence properties or limitations of the algorithm.
- **Clarify Novelty and Contributions:**
    - Clearly state the unique contributions of PROBS and how it advances the state of the art.

**Strengths And Weaknesses:**

**Strengths:**

1. **Innovative Approach with Beam Search:** The paper proposes an alternative to Monte Carlo Tree Search (MCTS) by integrating beam search with neural network training, potentially inspiring new research avenues in game-playing AI.
2. **Iterative Training Mechanism:** The method of iteratively refining value and policy networks using beam search outcomes is conceptually interesting and could contribute to the understanding of search and learning integration.
3. **Multi-Game Applicability:** By applying the algorithm to various games, the authors suggest that PROBS may generalize across different deterministic, perfect-information environments.

**Weaknesses:**

1. **Lack of Comparison with State-of-the-Art:** The paper does not compare PROBS with established algorithms like AlphaZero, making it difficult to assess its relative performance and contributions to the field.
2. **Insufficient Experimental Evaluation:** The evaluation is limited to weak baseline opponents, without testing against stronger, more sophisticated agents or in competitive settings.
3. **Omission of Key Related Work:** The paper lacks discussion of significant literature that combines search algorithms with neural networks, such as:
    - Anthony et al. (2017), "Thinking Fast and Slow with Deep Learning and Tree Search".
    - Guez et al. (2018), "Learning to Search with MCTSnets".
    This omission hinders the reader's understanding of how PROBS fits within the existing research landscape.
4. **Lack of Theoretical Justification:** The paper does not provide theoretical analysis explaining why PROBS should be effective or under what conditions it is expected to perform well.
5. **Unclear Novelty and Contributions:** The paper does not clearly articulate how PROBS differs from existing methods or what unique advantages it offers, leaving its contribution ambiguous.

---

### Review · Reviewer_8yzB · 2024-10-17

**Summary Of Contributions:**

This work considers the problem of game-playing for two-player, deterministic, perfect information settings. It posits that, instead of using Monte Carlo Tree Search as a general skeleton for this, the simpler beam search algorithm is also suitable.  An algorithm combining beam search and neural networks is proposed, which proceeds in an iterative fashion, and alternates steps of gathering data using the search informed by the neural networks and training the neural networks using this data. The algorithm is evaluated on several board games including Connect Four and a variant of Mancala, showing better performance than a few simple baselines.

**Audience:**

Yes

**Broader Impact Concerns:**

In my opinion, no ethical concerns are raised by this work.

**Claims And Evidence:**

No

**Requested Changes:**

### Major comments

M1. Please see the "Weaknesses" paragraph above, I consider this is essential for the work. Consider setting up the experimental comparison such that the number of node expansions is matched between beam search and MCTS, such that one can assess if the complexity of MCTS is "overkill" for these problems, as hinted to in the abstract and introduction.

M2. The proposed method follows what is, by now, a widely known and fairly successful recipe. There is very little novelty in either the proposed method or considered application. In its current form, the work does not meaningfully contribute to the state of the art.

M3. There is inherent randomness in the training process for PROBS. The presented evaluation ignores this, effectively presenting the results of a single run. Thus, it could be that the differences between the different methods that were observed are simply due to luck. This evaluation must be repeated, in my opinion, for all methods considering different random seeds, and reporting confidence intervals / error bars for all plots.

### Smaller comments

S1. The introduction gives too many details about how the method works but insufficient details about the research question(s) addressed, the contribution(s) of the paper, and its applications / implications. Consider moving some of these details to Section 3.

S2. The literature review is quite thin and pointers to highly relevant works are missing. For example, the iterative recipe featured in AlphaZero was proposed concurrently in [1]. Earlier works that use predictors for estimating the value function (such as [2]) or as a bonus when expanding the search tree (such as [3])  should also be surveyed.

S3. Beam search is a fairly standard algorithm and one would expect most readers in the TMLR audience to be familiar with it. For completeness, it can be listed in an appendix, but it probably does not belong in the main paper text.

S4. The term $V(s)$ (and analogously $Q(s,a)$) is used interchangeably to mean the true value function, an estimate of it, and a neural network. Consider differentiating between them e.g. by indicating estimates by $\hat{V}(s)$, and a neural net by $V_\phi(s)$ (the latter is used in places for $Q$).

S5. Consider describing all games used for evaluation at the beginning of Section 4 before discussing the experiments. It's also worth discussing the mechanics of the games and what differentiates them, this is arguably more important than detailing neural network input sizes.

S6. Section 6 definitely does not belong as the final section of the main text. These details can be provided either in either the beginning of Section 4 (before giving the results) or in an appendix.

### References

[1] Anthony, T., Tian, Z., & Barber, D. (2017). Thinking Fast and Slow with Deep Learning and Tree Search. In NeurIPS, 2017.

[2] Gelly, S., & Silver, D. (2007). Combining online and offline knowledge in UCT. In ICML, 2007.

[3] Rosin, C. D. (2011). Multi-armed bandits with episode context. Annals of Mathematics and Artificial Intelligence, 61(3), 203–230.

**Strengths And Weaknesses:**

**Strengths**. The basic hypothesis behind the work is sensible, the proposed method is simple and appears technically correct.

**Weaknesses**. A substantial weakness of the work, in my opinion, is the fact that its core hypothesis is not supported by sufficient evidence. The authors argue for replacing MCTS with a simpler search algorithm, but do not present experimental comparisons with MCTS. It is perfectly fine to not show comparisons with AlphaZero due to e.g. computational budget constraints, but one can nevertheless implement a baseline that follows the same structure as PROBS, replacing beam search with MCTS. Without this comparison, the key finding for the work seems to be "doing another kind of search also leads to reasonable results", which in my opinion, does not contribute meaningfully to the literature in this area.

---

### Review · Reviewer_3bkg · 2024-10-21

**Summary Of Contributions:**

he key contribution of this paper is the introduction of a new AlphaZero variant that replaces the traditional Monte Carlo Tree Search (MCTS) with beam search.

**Audience:**

Yes

**Claims And Evidence:**

No

**Requested Changes:**

The paper requires major revisions, see weaknesses.

**Strengths And Weaknesses:**

Strengths:

The paper introduces an interesting approach by employing beam search as an alternative to MCTS.

Weaknesses:

1.	Unclear motivation and contributions: The motivation for using beam search instead of MCTS is not well articulated. What advantages does beam search offer compared to MCTS?

2.	Limited novelty: The primary contribution is the replacement of MCTS with beam search, which I find lacks sufficient innovation to justify publication in TMLR.

3.	Inadequate experimentation: The paper does not include comparisons with essential baselines, such as AlphaZero and  simple board games like Connect Four, limiting the ability to evaluate the effectiveness of the proposed method.

---

### Decision · Action_Editor_pXmj · 2024-11-21

**Recommendation:** Reject

**Comment:**

The reviewers correctly point out issues, especially surrounding the experiments and the selection of baselines compared against. The authors have not engaged in any discussion to address these concerns.

**Audience:**

I think the paper is inside the scope in terms of audience, although it could be argued to be on the edge (considering the focus primarily on search, little focus on Machine Learning).

**Claims And Evidence:**

The claims are not sufficiently supported. Reviewers point out several issues, of which I think the most important one is the poor quality of baselines against which the proposed algorithm is evaluated. Only evaluating against 1-ply, 2-ply, and 3-ply searches is insufficient. At least baselines such as MCTS (which the paper explicitly mentions as potentially "replacing") should be tested.

**Resubmission Of Major Revision:**

The authors may consider submitting a major revision at a later time.